# Ascorbate as a Bioactive Compound in Cancer Therapy: The Old Classic Strikes Back

**DOI:** 10.3390/molecules27123818

**Published:** 2022-06-14

**Authors:** Jaime González-Montero, Silvia Chichiarelli, Margherita Eufemi, Fabio Altieri, Luciano Saso, Ramón Rodrigo

**Affiliations:** 1Basic and Clinical Oncology Department, Faculty of Medicine, Bradford Hill Clinical Research Center, University of Chile, Santiago 8380453, Chile; jagonzalez@uchile.cl; 2Department of Biochemical Sciences “A. Rossi Fanelli”, Sapienza University of Rome, 00185 Rome, Italy; silvia.chichiarelli@uniroma1.it (S.C.); margherita.eufemi@uniroma1.it (M.E.); fabio.altieri@uniroma1.it (F.A.); 3Department of Physiology and Pharmacology “Vittorio Erspamer”, Sapienza University of Rome, 00185 Rome, Italy; luciano.saso@uniroma1.it; 4Molecular and Clinical Pharmacology Program, Institute of Biomedical Sciences, Faculty of Medicine, University of Chile, Santiago 8380453, Chile

**Keywords:** cancer, ascorbate, oxidative stress, ferroptosis, iron

## Abstract

Cancer is a disease of high mortality, and its prevalence has increased steadily in the last few years. However, during the last decade, the development of modern chemotherapy schemes, new radiotherapy techniques, targeted therapies and immunotherapy has brought new hope in the treatment of these diseases. Unfortunately, cancer therapies are also associated with frequent and, sometimes, severe adverse events. Ascorbate (ascorbic acid or vitamin C) is a potent water-soluble antioxidant that is produced in most mammals but is not synthesised endogenously in humans, which lack enzymes for its synthesis. Ascorbate has antioxidant effects that correspond closely to the dose administered. Interestingly, this natural antioxidant induces oxidative stress when given intravenously at a high dose, a paradoxical effect due to its interactions with iron. Importantly, this deleterious property of ascorbate can result in increased cell death. Although, historically, ascorbate has been reported to exhibit anti-tumour properties, this effect has been questioned due to the lack of available mechanistic detail. Recently, new evidence has emerged implicating ferroptosis in several types of oxidative stress-mediated cell death, such as those associated with ischemia–reperfusion. This effect could be positively modulated by the interaction of iron and high ascorbate dosing, particularly in cell systems having a high mitotic index. In addition, it has been reported that ascorbate may behave as an adjuvant of favourable anti-tumour effects in cancer therapies such as radiotherapy, radio-chemotherapy, chemotherapy, immunotherapy, or even in monotherapy, as it facilitates tumour cell death through the generation of reactive oxygen species and ferroptosis. In this review, we provide evidence supporting the view that ascorbate should be revisited to develop novel, safe strategies in the treatment of cancer to achieve their application in human medicine.

## 1. Introduction

Cancer is a disease of high mortality whose incidence has risen progressively. Although multiple strategies have been developed in the last two decades for cancer prevention and treatment, such as targeted therapies and immunotherapy [1], mortality from cancer detected at advanced stages remains high [2]. Historically, the therapeutic antioxidant properties of ascorbate (also called ascorbic acid or vitamin C) have been studied in the context of cancer. However, since these studies were not replicated in clinical trials, the use of ascorbate was progressively relegated to a minor role. New knowledge about the pharmacokinetics of vitamin C, as well as some preclinical studies, have revived interest in treating cancer with high-dose vitamin C [3].

In the last decade, new mechanisms of action have been described for ascorbate at both the intracellular and extracellular levels. Its pharmacological profile has been the subject of intense study, which has led to the development of optimized strategies for the use of high-dose intravenous ascorbate to treat cancer. This new understanding of the pharmacological and molecular mechanisms of ascorbate in cancer has led to its consideration as a promising alternative treatment. 

### 1.1. Historical Aspects

Ascorbate first rose to prominence in association with the cause, treatment, and prevention of scurvy, a classical ascorbic acid deficiency disease now largely relegated to history [4]. Its first description dates to the times of Ancient Egypt, Greece, and Rome, where many people suffered from this vitamin C deficiency disease (Figure 1). In the Middle Ages, scurvy was an endemic disease in Northern Europe during winter, when fresh fruits and vegetables were unavailable [5]. 

### 1.2. Ascorbate in Human Medicine

Most applications of ascorbate in human medicine are related to its function in oxidation–reduction reactions. Ascorbate plays a pro-oxidant role by providing electrons to keep prosthetic metal ions in their reduced forms (e.g., cuprous ions in monooxygenases and ferrous ions in dioxygenases) [6].

Collagen production: Ascorbate plays the role of a coenzyme for prolyl and lysyl hydroxylases in order to convert protocollagen to collagen. Ascorbate is necessary for the maintenance of connective tissue and the wound healing process [7].Iron, haemoglobin metabolism, and erythrocyte maturation: Ascorbate enhances iron absorption by keeping it in the ferrous form. Due its reducing property, vitamin C helps the storage form of iron (complexed with ferritin) and its metabolisation [8]. Ascorbic acid is also involved in the production of the active form of folic acid and in erythrocyte maturation [9].Amino acid metabolism: Ascorbate is essential for tryptophan’s hydroxylation to hydroxytryptophan in the serotonin synthesis [10], and for the oxidation of p- hydroxyphenylpyruvate to homogentisic acid in tyrosine metabolism [11].Hormone synthesis: The synthesis of many hormones requires vitamin C. Ascorbate is an important cofactor of dopamine β-hydroxylase, the enzyme required to convert dopamine into norepinephrine [12]. Ascorbate is also an essential cofactor for the enzyme peptidylglycine α-amidating mono-oxygenase, which is required for the synthesis of vasopressin. Moreover, ascorbate may contribute to the magnitude of vasopressin biosynthesis [13]. Ascorbate is necessary for the hydroxylation reactions in the synthesis of corticosteroid hormones [14].Immunological function: Ascorbate enhances the synthesis of immunoglobulins and increases the phagocytic action of leucocytes [15]. Moreover, vitamin C has been shown to regulate the expression of pro-inflammatory and anti-inflammatory cytokines, to improve chemotaxis and phagocytosis, to enhance lymphocytic proliferation, and to assist in the oxidative neutrophilic killing of bacteria [13].Prevention of some diseases: Vitamin C concentrations may be low in acute illnesses, including myocardial infarction, pancreatitis, and sepsis [16]. Ascorbate, as an antioxidant, reduces coronary heart diseases and the risk of cancer [17]. Ascorbate has been shown to be involved in other biochemical activities [18], protecting the body from free radicals, enhancing the absorption of iron from vegetables, cereals, and fruits, helping in resistance against the common cold, and preventing some types of cancer [19].Ascorbate is widely known for its *immunological functions.* Ascorbate enhances the synthesis of immunoglobulins and increases the phagocytic action of leucocytes [15]. Moreover, vitamin C has been shown to regulate the expression of pro-inflammatory and anti-inflammatory cytokines, to improve chemotaxis and phagocytosis, to enhance lymphocytic proliferation, and to assist in the oxidative neutrophilic killing of bacteria [13]. More broadly, low levels of vitamin C have been implicated in a variety of acute illnesses, suggesting a potential application in *disease prevention.* Low levels of vitamin C have been described in association with acute myocardial infarction, pancreatitis, and sepsis [16]. Ascorbate, as an antioxidant, reduces coronary heart diseases and the risk of developing cancer [17]. Its myriad biochemical activities [18] have been proposed to contribute to a variety of health benefits, including protecting the body from free radicals, enhancing the absorption of iron from vegetables, cereals, and fruits, contributing to resistance against the common cold [19].

In this study, we review evidence supporting a role for ascorbate supplementation in the effective treatment of cancer, and highlight key points for its strategic development for a variety of applications in human medicine.

## 2. Pharmacology of Ascorbate

### 2.1. Ascorbate Distribution

Ascorbate is absorbed by the gut mucosa by simple diffusion and is aided in crossing the biological membrane by specific transport proteins. Efficient absorption occurs not only along the whole human intestine, but mainly in the duodenum [20]. In the intestine, ascorbate transporters are expressed both at the apical and basolateral epithelial membranes, but how this expression pattern along the intestinal tract is achieved is not yet known, though epigenetic regulation has been suggested [21]. 

### 2.2. Plasma Ascorbate Concentrations

In healthy individuals, tissue ascorbate levels range from around 0.5 to 10 mmoles/L [22], while plasma steady-state ascorbate levels are approximately 2.5-fold higher than those in tissue. Ascorbate is a potent, water-soluble, low-molecular-weight antioxidant capable of exerting multi-directional cellular effects. These effects are concentration-dependent and a pro-oxidant effect can be induced under elevated concentrations. Steady-state plasma vitamin C levels are tightly controlled at 70 to 85 µmol/L and peak values do not exceed 220 µmol/L, even with oral administration of doses as high as 3 g, six times daily [23]. In turn, vitamin C intravenously administered achieves plasma levels as high as 15,000 µmol/L [24]. It is of interest to note that the half-life of vitamin C is dependent on its plasma levels, being, at approximately 30 min, above 70 µmoles/L [24].

### 2.3. Ascorbate Pharmacodynamics and Pharmacokinetic

The complexity of ascorbate’s pharmacodynamics, combined with the multidimensionality of the human diseases in which ascorbate likely plays a beneficial role, has complicated the unambiguous demonstration of its relevance in human disease. However, recent studies of cellular ascorbate homeostasis will help to explain processes leading to ascorbate’s physiological and cellular distribution, which in turn has begun to bolster numerous experimental and clinical observations [25]. It has been suggested that the lack of fundamental knowledge about ascorbate’s biology has contributed to shortcomings in the design and interpretation of some published scientific studies [20]. Improved study design and better informed studies may now be expected given the new focus on well-performed studies of its fundamental biology. For instance, a recent molecular characterisation of tumour cells has suggested that the susceptibility of these cells to the therapeutic effect of ascorbate alone or in a combined therapy may be modulated by the overexpression or mutation of a suite of specific proteins [26], underscoring the benefits of multitarget therapy. Studies similar to this one have elevated the idea that the beneficial therapeutic properties of ascorbate may be augmented in the presence of other antineoplastic agents, thereby leading to a better clinical outcome and life quality and expectancy of patients. 

### 2.4. Ascorbate Transport 

Both ascorbate and dehydroascorbate (DHA) are absorbed in the human intestine, where most of the transport occurs in the reduced form. This absorption occurs in a sodium-dependent manner via a family of sodium-dependent vitamin C transporters (SVCTs) in the case of ascorbate, while DHA transport is mediated by facilitative glucose transporters (GLUTs), including sodium-independent facilitated diffusion [27]. Plasma levels of ascorbate increase with dietary/intravenous intake in a dose-dependent manner until levels plateau when these transporters became saturated. Interestingly, at high ascorbate levels, the expression of SVCTs is decreased. The high gradient concentrations established between cellular compartments are the result of the differential distribution of vitamin C transporters among tissues and organs for the different oxidised forms of the vitamin. Thus, the absorption, renal management, regulation of supply, distribution, excretion, and metabolism of vitamin C are highly dependent on the presence and abundance of widely distributed transporters. The high-affinity transporters are specific for ascorbate (SVCT-1 and SVCT-2), but the process is also linked to sodium and glucose transport [28]. It is worth noting that ascorbate transporters have been implicated in carcinogenesis [29]. This finding may help to explain the well-known observation that the plasma ascorbate levels of cancer patients are lower than those of healthy subjects [30]. Single-nucleotide polymorphisms influencing the coding regions of SLC23A1 and SLC23A2 of vitamin C transporters SVCT-1 and SVCT-2 were found to be sufficient to dramatically affect circulating levels of ascorbate [31]. Further studies about the role of ascorbate transporters may further uncover their relevance for modulating the plasma/tissue availability of ascorbate for redox reactions, which may represent a promising new direction for cancer prevention and treatment [29].

### 2.5. Subcellular Distribution 

The specific biological functions of ascorbate are influenced by its compartmentalisation within the eukaryotic cell [25], where its concentration and availability to regulate the redox state have been studied in detail in three cellular compartments: the endoplasmic reticulum, the mitochondrion, and the nucleus.

#### 2.5.1. Endoplasmic Reticulum

Ascorbate in this compartment plays a dual role, first as a local antioxidant towards reactive oxygen species (ROS) produced by “chemical accidents” [32], and, second, as a bioactive compound necessary for the maintenance of the redox state of ferrous iron at the active site of enzymes. Enzyme targets include the Fe(II)/2OG-dependent dioxygenases (e.g., prolyl hydroxylases and lysyl hydroxylases). Ascorbate also has pro-oxidant functions in the form of DHA in the ER [33]. The ascorbate can rapidly react with dithiols in unfolded or partially unfolded proteins in a PDI-dependent or independent manner [34], and ascorbic acid deficiency in auxotrophic species triggers ER stress [35]. 

#### 2.5.2. Mitochondrion

Mitochondrial ascorbate helps to maintain mitochondrial membrane potential, and it participates in ROS scavenging, which contributes to its anti-apoptotic effect. Ascorbate has been shown to protect mtDNA against ROS-induced mutations as 8-oxo-dGuanidine and apurinic/apyrimidinic sites [36]. Ascorbate has also been proposed to significantly attenuate the hydrogen peroxide-induced damage of mtDNA [37]. 

#### 2.5.3. Nucleus

The many published reports about ascorbate’s effects on cellular proliferation and differentiation may have been achieved in part through the modulation of gene expression in different cell types [38]. Ascorbate in the nucleus compartment can regulate enzymes catalysing epigenetically relevant reactions in the nucleus, such as Fe(II)/2OG-dependent dioxygenases, which catalyse the demethylation of histones and nucleic acids as well as the hydroxylation of specific histones. Moreover, several Fe(II)/2OG-dependent dioxygenases can also participate in DNA repair processes [39]. 

### 2.6. Ascorbate and Redox Balance

The strong reducing properties of ascorbate mainly derive from the two hydroxyl groups that are neighbours to the double bond in the lactone ring. The ascorbyl radical is a stable compound formed by the one-electron oxidation of ascorbate, whereas two-electrons oxidation forms dehydroascorbate (DHA), which has the same biological activity as ascorbate. DHA can also be reduced back to ascorbate in a reaction with reduced glutathione, catalysed by the enzyme dehydroascorbate reductase. In turn, DHA transport to the cell occurs via glucose transporters to be again reduced to ascorbate within the cell [40].

In contrast, ascorbate can create pro-oxidative conditions in the presence of transition metal ions, mainly copper, iron, and manganese, when ascorbate concentrations are within a millimolar concentration range. This interaction leads to increased ROS production via Fenton and Haber–Weiss reactions, particularly the production of reactive hydroxyl radical. In the Fenton reaction, ferrous ions react with hydrogen peroxide to produce hydroxyl radicals and hydroxyl anions.

### 2.7. Pro-Oxidant Effects of Ascorbate

Ascorbate, at plasma concentrations provided by a balanced diet, exerts an essential role as a nonenzymatic component of the antioxidant defence system. Indeed, the plasma ascorbate concentration is maintained at the micromolar level, even when high oral supplementation is administered. These physiological conditions are sufficient for the antioxidant roles exerted by ascorbate as an ROS scavenger and enzyme modulator against the development of oxidative stress. At millimolar concentrations, as well as in some pathological states, ascorbate acquires pro-oxidative activity, inducing DNA damage and cellular adenosine triphosphate depletion. This pro-oxidative effect depends mainly on the availability of free ionic iron. As shown in Figure 2, ascorbate is able to reduce Fe^3+^ to Fe^2+^, and the latter will react with oxygen through the Fenton reaction, thus forming ROS, mainly including the particularly reactive hydroxyl free radical. Ferrous iron resulting from the ascorbate reduction of Fe^3+^ easily reacts with oxygen, leading to the formation of reactive oxygen species and H_2_O_2_, which, in the presence of Fe^2+^, generates a highly reactive hydroxyl radical. 

Strikingly, significantly higher intracellular ascorbate concentrations can and must be obtained to promote these deleterious effects, with concentrations reportedly reaching up to 10 mM through active transport through cell membranes [41]. In tissues, the pituitary and adrenal glands achieve the highest ascorbate concentrations. Since the plasma ascorbate level is subjected to tight control, it cannot reach millimolar concentrations by oral administration. However, this control can be bypassed by intravenous administration. If higher plasma concentrations are achieved, then ascorbate autoxidation could give rise to the production of the ascorbyl free radical, radical anion superoxide, and H_2_O_2_, providing some clues for the therapeutic use of ascorbate [42]. Despite the fact that, at physiological conditions, the rates of these reactions are relatively slow, the presence of ionic iron or ions from other transition metals facilitates the production of hydroxyl radicals via the Fenton and Haber–Weiss reactions diagrammed above [43].

### 2.8. Ascorbate-GSH and Glucose Metabolism

A recent metabolomics study has suggested an important relationship between vitamin C and GSH in glucose metabolism, including glycolysis, the citric acid cycle (tricarboxylic acid; TCA cycle), and the pentose phosphate pathway. Levels of metabolites associated with upstream glycolysis, part of the TCA cycle (such as citrate and cis-aconitate), and the pentose phosphate pathway are increased in response to high concentrations of vitamin C, while levels of metabolites downstream of glycolysis are decreased (except for citrate and cis-aconitate). The authors attributed this finding to GAPDH inactivation by vitamin C-induced oxidative stress. They found that upstream metabolites of GAPDH accumulated, whereas downstream metabolites of GAPDH were depleted, and ATP concentrations were decreased in response to high-dose vitamin C. The authors further suggested that vitamin C-mediated oxidative stress induces the depletion of NADH, which inhibits glycolytic flux, ultimately resulting in decreased ATP levels due to the inhibition of energy metabolism, causing cell death [44].

Vitamin C is involved in a variety of oxidative mechanisms in glucose metabolism. High levels of vitamin C induce high ROS levels and the oxidation of GSH, which also plays a significant role in cellular defence against oxidative stress. Accompanied by the direct glutathionylation of GAPDH in glycolysis, glucose metabolism was altered by vitamin C treatment. Furthermore, the reduced glutathione ratio triggered by vitamin C resulted in altered GSH metabolism via de novo synthesis.

Other recent studies have found that serum concentrations of GSH are associated with various disease conditions [45], including cancer and neurodegenerative disease susceptibility. Vitamin C stimulates the oxidation of GSH towards dimerisation in oxidised form (GSSG) and leads to the accumulation of hydrogen peroxide (H_2_O_2_), leading to apoptosis.

### 2.9. Ascorbate and Enhancement of Cell Death

The propensity of high concentrations of ascorbate to induce oxidative injury suggests its consideration for pharmacological use, particularly for targeting cells with increased ROS vulnerability, such as malignant cancer cells [46]. Selective killing of tumour cells by ascorbate has been demonstrated in several models [47]. Cancer cells are characterised by already increased levels of steady-state ROS, making them more easily undergo apoptosis with additional oxidative injury [27]. As oral ascorbate did not demonstrate effectiveness in reducing the cancer burden in clinical trials, it has not been further pursued in conventional oncology settings. In contrast, intravenous ascorbate produces millimolar plasma levels, leading to cancer cell death, without adversely affecting normal tissues. Intravenous ascorbate combined with carboplatin and paclitaxel inhibited ovarian cancer in a murine model, and also reduced the associated toxicity in patients with ovarian cancer [48]. An in vitro study evaluated the effects of millimolar ascorbate concentrations on the cell viability of four canine melanoma cell lines and found that these melanoma cell lines had reduced viability and exhibited concentration-dependent ascorbate-induced apoptosis [49].

A separate study on the sensitivity of human lymphoma cells to ascorbate reported that extracellular, but not intracellular, ascorbate mediated cell death, in a strictly H_2_O_2_-dependent manner [50]. In turn, H_2_O_2_ generation was dependent on the ascorbate concentration and incubation time, and displayed a linear relationship with ascorbate radical formation. Therefore, the authors conclude that ascorbate at concentrations achieved only by intravenous administration may be a pro-drug for the formation of H_2_O_2_, and that blood can be a delivery system of the pro-drug to tissues, thus giving plausibility to intravenous ascorbate use in cancer treatment [50].

### 2.10. Ascorbate and Ferroptosis

During the last decade, ferroptosis, a form of non-apoptotic regulated cell death, has been linked with the occurrence of oxidative stress and iron overload, in a close relationship with cancer initiation, progression, and suppression [51]. Although ferroptosis has numerous regulatory inputs, oxidative stress induced by therapeutic ascorbate represents an exciting potential target in cancer therapy. As previously discussed, ascorbate causes a biphasic response, causing antioxidant effects at micromolar levels and a pro-oxidant effect at millimolar concentrations. Studies on cell lines reported an EC50 value for ascorbate of around 0.5 mM, while ascorbate concentrations above 1.0 mM decreased the viability of cells to practically zero after 24 h of treatment [52]. These findings are in line with a report about fibroblasts exposed to ascorbate in a model of simulated ischemia–reperfusion [53].

Redox intracellular balance is a critical determinant of cellular vulnerability to further oxidative challenge. This predicts that cancer cells, which have increased intracellular ROS concentrations at steady state, should be damaged or die under oxidative conditions that do not adversely affect healthy cells at physiologic redox states. Iron homeostasis is dysregulated in the induction of ferroptosis, as iron-independent ROS in the cell induce ferritinophagy and can increase intracellular iron levels [54]. The level of labile iron from intracellular pools is ultimately the chief culprit in the generation of ROS through the Fenton reaction [55]. In addition, ferroptotic cells undergo inactivation of glutathione peroxidase 4 (GPX4), thereby depleting cells of reduced glutathione, one of the most important endogenous antioxidants [56]. GSH is an essential substrate for GPX4 and exerts antiferroptotic activity, as GPX4 transforms toxic lipid peroxides into non-toxic fatty alcohols and prevents membrane phospholipids from reacting with ROS to produce lipid peroxides that directly induce ferroptosis [57]. The antioxidant system comprising the GPX4 core is considered one of the most powerful cellular protections from ferroptosis. Therefore, conditions impairing this enzyme function in all cells increase their vulnerability to ferroptosis, which has particular strategic relevance for anti-tumour therapy design [58,59]. A recently discovered ferroptosis inducer, FINO2, inhibited GPX4 while directly oxidising polyunsaturated fatty acids, providing yet another angle for prospective intervention in ferroptosis by adjusting GPX4 levels [60].

Lastly, the tumour suppressor gene p53 can prevent lethal effects under low to moderate ROS levels, representing another ROS-sensitive player in ferroptosis control. In contrast, at high ROS levels, p53 causes ferroptosis by inhibiting SLC7A11 gene transcription [61]. Parallel antioxidant mechanisms in mitochondria, coordinated by dihydroorotate dehydrogenase [62], contribute to ferroptosis resistance.

### 2.11. The Ascorbate Paradox: Biphasic Effect

Ascorbate has a double-faced character, as it exhibits a pro-oxidant activity arising from its known antioxidant property depending on its extracellular concentration. It can generate reactive free radicals, which induce cytotoxic effects at pharmacologic concentrations [63].

Endogenous vitamin C is an antioxidant at normal physiological concentrations, which, in human plasma, is 40–80 µM [64]. However, ascorbate can donate an electron to redox-active transition metal ions, such as cupric (Cu^2+^) or ferric (Fe^3+^) ions, reducing them to cuprous (Cu^+^) and ferrous (Fe^2+^) ions, respectively [65]. Reactions of ascorbic acid with metals such as Cu^2+^ and free iron in vitro are thought to lead to the production of H_2_O_2_ [65] (Figure 2). Thus, the biphasic effect of ascorbate is based on its metal-dependent ability to react with H_2_O_2_ and reduced metal ions, such as iron or copper, to produce hydroxyl radicals by Fenton chemistry. In this context, ascorbate can cause strand breakage in DNA in the presence of oxygen and can initiate cell death in malignant tissue culture, possibly through the generation of H_2_O_2_ [66]. Thus, the capacity of ascorbic acid to act as a pro-oxidant molecule depends on the redox potential of the cellular environment, the presence or absence of transition metals, and the local concentration of ascorbate [8]. In recent years, it became evident that pharmacological high-concentration ascorbate in the millimolar range bears selective cytotoxic and pro-oxidant effects on cancer cells in vitro and in vivo [67]. The optimum concentration of ascorbic acid used to produce potential pro-oxidant-associated cytotoxic effects was found to be 3–5 mM in vitro, 0.88–5 mM in vivo in animals, and 0.5–2 mM ex vivo in humans [63]. This effect is dose-dependent, catalysed by serum components and mediated by reactive oxygen species and ascorbyl radicals. Thus, ascorbate either acts as a pro-oxidative pro-drug that catalyses hydrogen peroxide production instead of acting as a radical scavenger [64] or inhibits the antioxidant systems in the presence of iron, which in turn induces lipid peroxidation, depending on the concentration gradient and redox state of a cell [68].

At low millimolar concentrations in vivo, ascorbate generates superoxide radicals, hydrogen peroxide, and extracellular ascorbyl, which drives its cytotoxic activity; however, concentrations as high as 20 mM did not pose any risk to the lineage of non-malignant cells [64]. Normal tissue has rich and adequate blood flow and contains antioxidant enzymes, such as catalase and glutathione peroxidase, permitting the destruction of any hydrogen peroxide formed, and in this way preventing damage by ROS [68] and preventing cellular damage via pro-oxidant ascorbic acid. On the other hand, malignant cell tissue is characterised by abnormal blood flow and fewer antioxidant enzymes, allowing for the formation of hydrogen peroxide and increased cellular damage [68]. It was demonstrated that concentrations well above physiological plasma concentrations (40–80 μm in plasma), e.g., between 1 mM and 10 mM, are toxic for neoplastic cells in vitro—for example, for melanoma and neuroblastoma cells, where concentrations from 10 nM to 1 mM can induce apoptosis probably due to the pro-oxidant capacity of ascorbate and its biphasic effect [64]. Similar bimodal effects have been suggested in the specific distinction and targeting of oral neoplasms, while leaving the healthy oral environment intact [64]. Figure 3 shows a scheme of the possible effects of a high concentration of ascorbate on tumour cells.

### 2.12. Hormetic Effect of Ascorbate

L-ascorbate was first identified as a potential therapeutic agent against cancer in the 1970s, and a study that provided oral administration of 10 g per day of dietary ascorbate increased the average survival time in a cohort of cancer patients [69]. Subsequent well-controlled studies failed to recapitulate this striking result [70]. A recent hypothesis has attributed this discrepancy to a hormetic effect derived from the cellular uptake of ascorbate. Hormesis is an adaptive response of cells and organisms to an environmental agent, characterized by a biphasic dose–response relationship consisting of low-dose stimulation or a beneficial effect and a high-dose inhibitory or toxic effect, although the issue of beneficial/toxic effects is reserved for the evaluation of the biological context of the response [71]. Up-regulation of sodium-dependent vitamin C transporter 2 (SVCT-2) has been suggested as an indicator for the increased L-ascorbic acid uptake in cancer cells [72]. More recently, a study demonstrated an hormetic response in high SVCT-2-expressing cell lines, proposing that L-ascorbic acid has a biphasic role in cancer cells, depending on ROS generation. High ascorbate concentrations generate ROS sufficiently to act as an anti-cancer agent, regardless of SVCT-2 expression, and a hormetic proliferation response was found, with low SVCT-2-expressing cell lines having sufficient ascorbate uptake to generate ROS. Thus, this study demonstrated that ascorbate could be an effective chemotherapeutic agent, with sufficient delivery into colorectal cancer cell lines, that showed cell growth inhibition and apoptosis [73]. Similarly, in vivo studies in an Apc/KrasG12D mutant mouse model of intestinal cancer indicated that high-concentration vitamin C can impair tumour growth. Although it is unclear whether a similarly beneficial response will occur in human tumours, these data provide a mechanistic rationale for exploring the therapeutic use of ascorbate [74].

## 3. Application of Ascorbate in Cancer Therapy

### 3.1. Experimental Models

The role of ascorbate has been studied in both animal and in vitro models, both for its cytotoxic effect as a monotherapy treatment and as an adjuvant to oncologic therapies. Ascorbate been shown to have a cytotoxic effect in a colorectal cancer cell line, where its effect is dependent on the expression of sodium-dependent vitamin C transporter 2 (SVCT-2) [27,73]. In gastric cancer cell lines, the sensitivity of cells to the cytotoxic effect of ascorbate was inversely correlated with GLUT-1 expression, suggesting GLUT-1 expression as a biomarker predicting sensitivity to ascorbate therapy [75]. Direct cytotoxic effects of ascorbate were also demonstrated in a murine model of melanoma [76].

As an adjuvant to conventional chemotherapy treatments, ascorbate has been shown in non-small-cell lung cancer cell lines to enhance the cytotoxicity of chemotherapy [77,78]. In pancreatic cancer cell lines, ascorbate enhances the cytotoxic effect of gemcitabine and paclitaxel by decreasing chemoresistance [79]. In a study that included in vitro experiments with 11 different cancer cell lines, around half of the cell lines tested were resistant to ascorbate cytotoxicity, a response associated with high levels of catalase activity [80], suggesting a potential role for catalase in mediating ascorbate’s cytotoxicity effects. Finally, both cell lines and a murine xenograft model of colorectal cancer with KRAS mutation demonstrated that high concentrations of ascorbate enhanced the cytotoxic effect of chemotherapy [81]. In parallel, in preclinical models of KRAS-mutated colorectal cancer, the combination of ascorbate and chemotherapy improved tumour regression, and this response depended on SVCT-2 expression in tumour cells [82].

More recently, ascorbate has also been powerfully combined with biological therapy in cancer treatment. In HER2-positive breast cancer cell lines, treatment with high concentrations of ascorbate and a monoclonal antibody targeting HER2, trastuzumab, resulted in a decrease in tumour cell proliferation compared to trastuzumab alone [83]. Furthermore, a combination of a high concentration of ascorbate and immunotherapy (anti PD1 and/or anti CTLA4 antibodies; Table 1) showed improved cytotoxic effects in pancreatic, breast, melanoma, and colorectal cancer models [84]. These promising results suggest that ascorbate enhances cytotoxic tumour cell killing in immunotherapy. Further studies addressing the mechanisms of this phenomenon will make clear whether this synergy will be clinically relevant.

### 3.2. Clinical Models

The rationale for using ascorbate therapy in cancer was first provided by the pioneering studies of Linus Pauling in the 1970s and 1980s, which suggested that high doses of ascorbate (10 g/day intravenously for 10 days followed by 10 g/day orally indefinitely) are useful in the treatment of cancer, improving overall survival and quality of life [69]. Two subsequent, independent, randomised clinical trials showed no benefit for survival with the use of oral ascorbate [70,95]. In the 2000s, additional limitations on oral ascorbate uptake were discovered, leading to the recognition that the route of administration of ascorbate is critical for achieving high concentrations in tissue [23]. The following decade, significant progress was made in understanding the cellular effects of ascorbate, finding that ascorbate drives the re-expression of tumour suppressor genes and inhibition of the epithelial–mesenchymal transition, among other effects [96], establishing the modern principles for the use of intravenous ascorbate in high doses.

#### 3.2.1. Cancer Prevention

Overall, observational prospective cohort studies have found no association or a modest inverse association between ascorbate intake and the risk of cancer [97,98,99,100]. 

In breast cancer, two large prospective studies found that dietary intake of ascorbate is inversely associated with breast cancer incidence in certain subgroups. In one study (the Nurses’ Health Study), premenopausal women with a family history of breast cancer who consumed an average of 205 mg/day of ascorbate from food had a lower risk of breast cancer than those who consumed an average of 70 mg/day [101]. In a second study (referred to as the Swedish Mammography Cohort), overweight women who consumed an average of 110 mg/day of ascorbate had a lower risk of breast cancer compared with overweight women who consumed an average of 31 mg/day [102]. More recent prospective cohort studies reported no association between dietary or supplemental ascorbate oral intake and breast cancer [103,104].

In stomach cancer, no association was observed between dietary intake of vitamin C and gastric cancer risk [105]. A case–control study of the European Prospective Study on Diet and Cancer (EPIC) found a 45% lower risk of gastric cancer incidence in individuals in the highest quartile (≥51 µmol/L) versus the lowest (<29 µmol/L) of plasma vitamin C concentration.

In colorectal cancer, data pooled from 13 prospective cohort studies determined that dietary intake of ascorbate was associated with a 19% lower risk of colon cancer [106]. However, each of the cohort studies used self-administered food frequency questionnaires to assess ascorbate intake, and other healthy behaviours and/or folate intake may have confounded the association. Later, the Physicians’ Health Study II, a randomised, placebo-controlled trial, found no effect of vitamin E and ascorbate intake (500 mg/day) on overall cancer risk for prostate, bladder, or pancreatic cancer, though there was a marginal reduction in colorectal cancer, compared to a placebo [107].

#### 3.2.2. Cancer Treatment

Table 2 summarises observational studies and clinical trials that have tested the use of intravenous ascorbate as a cancer treatment. Overall, current evidence for the efficacy of intravenous ascorbate in cancer patients is limited to observational studies, uncontrolled interventional studies, and case reports [108,109]. Larger and longer clinical trials are needed to assess the efficacy of intravenous ascorbate on cancer progression and overall survival.

Ascorbate has been used to promote damage to cancer cells in radiotherapy and chemotherapy through oxidative stress generation. Ascorbate has been shown to have cytotoxic effects as a therapy adjuvant to chemoradiation in the treatment of oesophageal cancer [116] and gastric cancer [117], non-small-cell lung cancer and glioblastoma [78], and pancreatic cancer [118,119]. In addition, intravenous ascorbate has been found to mitigate damage to normal tissue following chemoradiation therapy [91,120], and it may also have synergistic effects with palliative radiotherapy in patients with bone metastases [121].

The effect of ascorbate has been proven as an adjuvant therapy to chemotherapy, as well. Several studies, including case reports and clinical trials, have studied the effect of IV ascorbate in patients with different types of cancer. Two initial reports showed that high-dose IV ascorbate treatment is well tolerated in cancer patients [122,123]. However, one study with only three cases showed long survival times of patients [123], while the second study, reporting 24 cases, failed to detect any anti-cancer activity of ascorbate [122]. In a study with 60 patients with different types of cancer, IV ascorbate improved quality of life [124]. Ascorbate administrated alone also improved quality of life in a study including 17 patients with different solid tumours, although no patient showed an objective anti-tumour response [110]. Two additional studies evaluated the effect of IV ascorbate on the survival of patients with stage IV pancreatic cancer receiving standard chemotherapy treatment. Both studies reported a reduction in tumour mass and improvements in overall survival [111,112].

In a randomised controlled trial on 27 patients with ovarian cancer, IV ascorbate treatment reduced chemotherapy-associated toxicity; however, a minimal effect on patient survival was observed [48]. Similarly, none of the 23 patients with metastatic castration-resistant prostate cancer that received IV ascorbate experienced disease remission or anti-cancer effects [115]. In contrast, 13 glioblastoma patients receiving radiotherapy and 14 non-small-cell lung cancer patients receiving chemotherapy showed improved overall survival with IV ascorbate treatment [78]. Similarly, a report analysing 14 pancreatic cancer patients receiving IV ascorbate and chemotherapy showed improved survival [114]. Finally, in a phase I clinical trial for metastatic colorectal and gastric cancer, 36 patients received high-dose ascorbate in combination with mFOLFOX6 or FOLFIRI. Potential clinical efficacy was observed in these patients [125]. One recent study [84] reported that ascorbate may maximise the anti-proliferative effect of immunotherapy. Further studies must be done to determine whether this combination may be more efficacious in reducing cancer burden than combination with more traditional therapies suggests.

In conclusion, most studies to date have reported that ascorbate treatment supports improved quality of life or reduced chemotherapy-related side effects, but few of these reports have demonstrated a clear anti-tumour effect or benefits in overall survival with ascorbate treatment, especially when it is applied as an adjuvant therapy. The differential results could be due to the disparities between the studies, such as doses, number of patients, types of cancer, methodologies, and parameters studied, an issue that has been previously analysed by several reviews [108,109,126,127].

## 4. Conclusions

Ascorbate has been widely studied in human health and disease. After initial promising reports, the effect of ascorbate in cancer treatment has been revisited in recent decades. Although ascorbate appears to provide anti-tumour effects in animal models of cancer and in cell lines, this benefit has yet to be realised in clinical trials or in clinical settings. However, a recent revival of interest in ascorbate as a potential adjuvant therapy to conventional cancer therapies (e.g., chemotherapy, radiotherapy, or chemoradiotherapy) has also revived interest in the study and use of this therapy in cancer treatment. Today, we have a better understanding of the biology and pharmacology of ascorbate, particularly of the differences between oral and intravenous administration, and robust experimental evidence in model systems and cell lines suggests the potentially powerful anti-tumour effects of ascorbate when administered at high doses. Taken together, we suggest that future studies in cancer patients should focus on rational interventions that utilise high doses of intravenous ascorbate, both as an adjuvant to standard therapies and as a monotherapy.

## Figures and Tables

**Figure 1 molecules-27-03818-f001:**
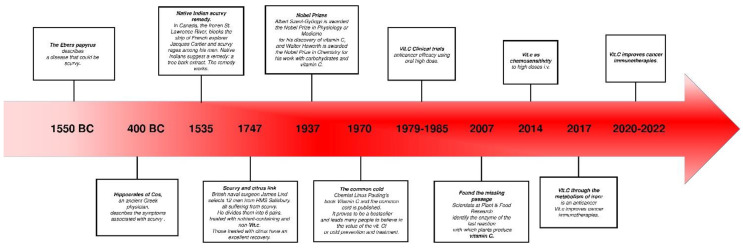
Timeline summarising the various uses of ascorbate in human medicine throughout history.

**Figure 2 molecules-27-03818-f002:**
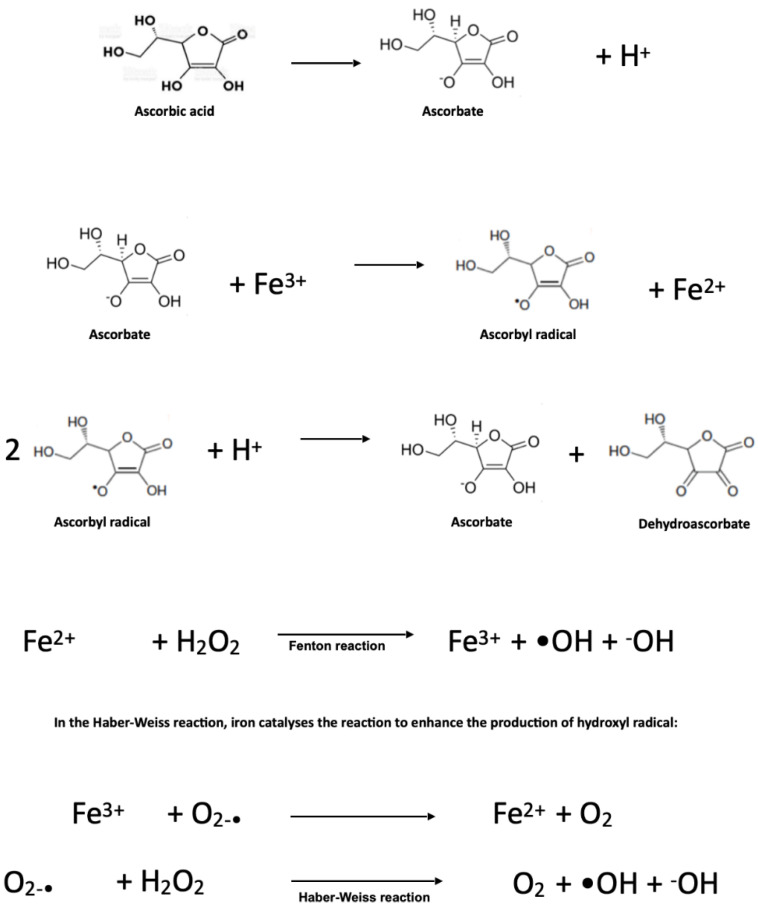
Pro-oxidant effect of ascorbate when it interacts with free iron.

**Figure 3 molecules-27-03818-f003:**
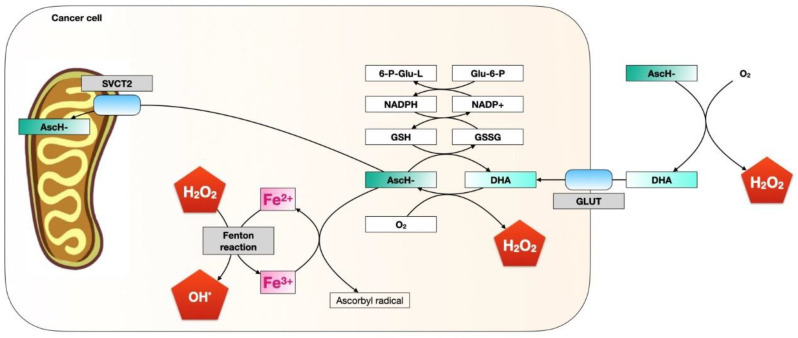
Scheme of the possible effect of high doses of ascorbate on tumour cells. AscH-: High-dose ascorbate. DHA: Dehydroascorbic acid.

**Table 1 molecules-27-03818-t001:** Summary of the most important in vivo and in vitro studies that have studied the effect of ascorbate in combination with systemic oncological therapies in the treatment of cancer.

Type Drug	Cancer Type	Study Design	Ascorbate Concentration or Dose	Main Findings	Reference
Radiotherapy	Pancreatic cancer	In vitro study. *n* = 1 cell line. Concomitant radiotherapy 2 Gy.	4 mM during 24 h	Radio-sensitising effect of ascorbate	[85]
Fluorouracil	Colorectal and gastric cancer	In vitro and in vivo study. Cell lines of colorectal and gastric cancer. *n* = 60 mice.	1 mM in vitro 4 g/kg intra peritoneal in vivo	In vitro synergy enhanced efficacy of chemotherapy	[79,86]
Anti-PD-1 and Anti-CTL-4	Breast, colorectal, and pancreatic cancer	In vivo study. *n* = 13 immunocompetent syngeneic mice.	4 g/kg intraperitoneal	Synergy and effective anti-tumour immune memory	[84]
Carboplatin	Gastric cancer	In vitro and in vivo study. *n* = 2 cell line. *n* = 60 athymic-nu/nu mice.	1 mM in vitro 4 g/kg intraperitoneal in vivo	Enhanced efficacy	[79]
Cetuximab	Colorectal cancer with KRAS mutation	In vitro study. *n* = 5 cell lines.	0.3, 0.5, and 0.7 mM	Synergy and abrogates resistance	[82]
Cisplatin	Gastric, cervical, oral squamous, and ovarian	In vitro studies.	Ranging from 0.0002 mM to 2 mM	Synergy enhanced efficacy	[87,88]
Doxorrubicin	Cervical cancer	In vitro. *n* = 2 cell lines.	1.25, 3.3, and 16 mM	Synergy	[89]
Etoposide Temozolamide	Glioblastoma multiforme	In vitro. *n* = 1 cell line.	1 mM	Enhanced efficacy	[90]
Gemcitabine	Pancreatic cancer	In vitro, in vivo. *n* = 6 cell lines. *n* = 32 mice.	0.001 mM in vitro 4 g/kg intraperitoneal in vivo	Enhanced efficacy	[91]
Irinotecan Oxaliplatin	Colorectal and gastric cancer	In vitro and in vivo studies.	0.15–13.3 mM in vitro 4 g/kg intraperitoneal in vivo	Synergy in vitro enhanced efficacy	[86,92]
Paclitaxel	Oral squamous and gastric cancer	In vivo and in vitro studies. *n* = 2 cell lines. *n* = 60 mice.	1 mM in vitro 4 g/kg intraperitoneal	Enhanced efficacy	[79,93]
Vermurafenib	BRAF mutant melanoma	In vitro and in vivo study. *n* = 2 cell lines. *n* = 18 c57BL/6 mice.	1.5 mM in vitro 0.03 mg/kg oral	Synergy and abrogates resistance	[94]

**Table 2 molecules-27-03818-t002:** Summary of observational studies and clinical trials that have tested the use of intravenous ascorbate as a cancer treatment.

Study Characteristics	Ascorbate Dose	Results	Reference
Observational studies			
IV ascorbate in advanced tumours	*n* = 17 patients treated with ascorbate in dose: 30, 50, 70, 90, and 110 g/m^2^ for 4 consecutive days for 4 weeks.	3 patients had stable disease, 13 had progressive disease.	[110]
IV ascorbate in advanced pancreatic adenocarcinoma	*n* = 11 patients treated with IV ascorbate ranging 15–125 g twice weekly, with gemcitabine.	Mean plasma ascorbate levels were significantly higher than baseline. Mean survival time of subjects completing 8 weeks of therapy was 13 ± 2 months.	[111]
IV ascorbate in pancreatic adenocarcinoma stage IV	*n* = 14. 50, 75, and 100 g per infusion (3 cohorts) thrice weekly for 8 weeks. Concurrent therapy with gemcitabine and erlotinib.	50% of patients had stable disease. Survival analysis excluded 5 patients who progressed quickly (3 died). Overall mean survival was 182 days.	[112]
Stage III and IV serous ovarian cancer	*n* = 25. 75–100 g IV ascorbate twice weekly for 12 months (target plasma concentrations 20–23 mM).	8.7 month increase in progressive-free survival in ascorbate-treated arm.	[48]
Various cancer types (lung, rectum, colon, bladder, ovary, cervix, tonsil, breast, biliary tract)	*n* = 16. 1.5 g/kg body weight infused for three times	Patients experienced stable disease, increased QOL, and functional improvement.	[113]
Glioblastoma under treatment with chemoradiation with concomitant temozolamide	*n* = 13. Radiation phase: radiation (61.2 Gy in 34 fractions), temozolamide, and ascorbate (ranging from 15 to 125 g, 3 times per week for 7 weeks). Adjuvant phase: ascorbate (2 times per week, dose escalation until 20 mM plasma concentration, around 85 g infusion).	Progression-free survival 13.3 months. Overall survival 21.5 months.	[78]
Advanced stage non-small-cell lung cancer	*n* = 14. 1 cycle is 21 days. IV carboplatin, IV paclitaxel, and IV pharmacological ascorbate (two 75 g infusions per week, up to 4 cycles).	Partial responses (*n* = 4) and stable disease (*n* = 9), disease progression.	[78]
Locally advanced or metastatic prostate cancer	*n* = 14. IV ascorbate 25–100 g. Concurrent chemotherapy: Gemcitabine.	Patients experienced a mix of stable disease, partial response, and disease progression.	[114]
Castration-resistant prostate cancer	*n* = 23. 5 g weekly during week 1, 30 g weekly during week 2, and 60 g weekly during weeks 3–12.	Adverse events were thought to be more likely related to disease progression than ascorbic acid.	[115]
**Phase I clinical trials**			
Locally advanced pancreatic cancer treated with chemoradiation	IV ascorbate concomitant to radiation and gemcitabine. Ascorbate dosing: 50 g administered intravenously (by IV) during radiation therapy, for approximately 5 to 6 weeks.	US National Library of Medicine. https://clinicaltrials.gov/ct2/show/ NCT01852890 (2018). Accessed on 1 May 2022.	On going
Glioblastoma multiforme treated with chemoradiation (temozolamide)	IV ascorbate concomitant to radiation and temozolamide. Ascorbate dosing: 15, 25, 50, 62.5, 75, and 87.5 g administered by IV three times a week until 1 month after radiation is completed (approximately 12 weeks).	US National Library of Medicine. https://clinicaltrials.gov/ct2/show/ NCT01752491. Accessed on 1 May 2022.	On going
Metastatic pancreatic cancer treated with gemcitabine and nab-paclitaxel	The dose level for phase II patients will be determined following completion of the phase 1b study based on response from 3–6 patients receiving the designated dose level of ascorbic acid.	US National Library of Medicine. https://clinicaltrials.gov/ct2/show/ NCT03797443. Accessed on 1 May 2022.	On going
**Phase II clinical trials**			
Stage IV non-small-cell lung cancer patients treated with chemotherapy	IV ascorbate: 75 g per infusion, two infusions per week (each 3 weeks) for 4 cycles.	US National Library of Medicine. https://clinicaltrials.gov/ct2/show/ NCT02420314. Accessed on 1 May 2022.	On going
Pharmacological ascorbate combined with radiation and temozolamide in glioblastoma multiforme: a phase II trial	Intravenous infusions of ascorbate of 87.5 g administered three times weekly during chemoradiation. After radiation, ascorbate is administered twice weekly through the end of cycle 6 of temozolomide.	US National Library of Medicine. https://clinicaltrials.gov/ct2/show/ NCT02344355. Accessed on 1 May 2022.	On going
Various solid tumour malignancies (colorectal, pancreatic, and lung cancer)	IV ascorbate: 1.25 g/kg for 4 days per week for 2–4 consecutive weeks or up to 6 months.	US National Library of Medicine. https://clinicaltrials.gov/ct2/show/ NCT03146962. Accessed on 1 May 2022.	On going
Pharmacological ascorbate with concurrent chemotherapy and radiation therapy for non-small-cell lung cancer	IV ascorbate dosing: 75 g per infusion. 3 infusion per calendar week.	US National Library of Medicine. https://clinicaltrials.gov/ct2/show/ NCT02905591. Accessed on 1 May 2022.	On going
Pharmacological ascorbate, gemcitabine, nab-paclitaxel for metastatic pancreatic cancer	IV dosing: 75 g of ascorbate 3 times per calendar week for each week of the chemotherapy cycle.	US National Library of Medicine. https://clinicaltrials.gov/ct2/show/ NCT02905578. Accessed on 1 May 2022.	On going
Ascorbic acid in combination with docetaxel in men with metastatic prostate cancer	Patients receive docetaxel IV on day 1 and ascorbic acid IV twice weekly. The first ascorbic acid treatment will be given on day 1 (same day as docetaxel). Treatment repeats every 21 days for 8 courses in the absence of disease progression or unacceptable toxicity.	US National Library of Medicine. https://clinicaltrials.gov/ct2/show/ NCT02516670. Accessed on 1 May 2022.	On going
**Phase III clinical trials**			
IV ascorbic acid in advanced gastric cancer	In patients treated with chemotherapy, ascorbate IV 20 g day (days 1–3) will be administered every 2 weeks.	US National Library of Medicine. https://clinicaltrials.gov/ct2/show/ NCT03015675. Accessed on 1 May 2022.	On going
Stage IV colorectal cancer	IV ascorbic acid (1.5 g/kg/day, days 1–3, every 2 weeks) in combination with FOLFOX and bevacizumab versus treatment with FOLFOX and bevacizumab alone as first-line therapy for advanced colorectal cancer.	US National Library of Medicine. https://clinicaltrials.gov/ct2/show/ NCT02969681. Accessed on 1 May 2022.	On going

## Data Availability

Not aplicable.

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
