# Peer review of "Ascorbate as a Bioactive Compound in Cancer Therapy: The Old Classic Strikes Back"

_molecules, 2022, doi:10.3390/molecules27123818_

Round 1
Reviewer 1 Report
I have gone through the manuscript "Ascorbate as a bioactive compound in cancer therapy: The old classic strikes back" authored by Montero et al. The review is well written and providing novelty to already existing knowledge and literature. In my view manuscript can be accepted after some minor revisions related to syntax, language, and grammar.
Abstract should be concise and crisp. It is lengthy with so many language flaws.
Moreover, manuscript lacks the methodology part like exclusion and inclusion criteria of related literature that has been used for collection of data.
Errors related to language, syntax and grammar are so many some examples are as follow:
Line 16 - “serious disease “ should be omitted
Line 17, 18 – use of “and” so many times is breaking the flow.
Line 24- “high dosing” should be high dose.
Line 96-104- use of “helping” so many times.
Line 145- sentence should contain “either-or” . If it means both types of mechanism then it should be written using “-and”
Conclusion could be elaborated as it is simply the replicate of abstract.
Author Response
Reviewer 1: I have gone through the manuscript "Ascorbate as a bioactive compound in cancer therapy: The old classic strikes back" authored by Montero et al. The review is well written and providing novelty to already existing knowledge and literature. In my view manuscript can be accepted after some minor revisions related to syntax, language, and grammar.
- Abstract should be concise and crisp. It is lengthy with so many language flaws. Moreover, manuscript lacks the methodology part like exclusion and inclusion criteria of related literature that has been used for collection of data.
- Response: The present versión of the manuscript provides a concise Abstract accounting for the need to study a novel role of ascorbate in the therapeutic strategies as an adjuvant novel tool. Language flaws were corrected. Inclusion criteria of related literature has been focused on the conditions leading to the induction of oxidative stress achieved by intravenously high dose ascorbate.
- Errors related to language, syntax and grammar are so many some examples are as follow: Line 16 - “serious disease “ should be omitted
- Response: The content indicated was corrected and indicated in the text with track changes.
- Line 17, 18 – use of “and” so many times is breaking the flow.
- Response: The content indicated was corrected and indicated in the text with track changes.
- Line 24- “high dosing” should be high dose.
- Response: The content indicated was corrected and indicated in the text with track changes.
- Line 96-104- use of “helping” so many times.
- Response: The content indicated was corrected and indicated in the text with track changes.
- Line 145- sentence should contain “either-or” . If it means both types of mechanism then it should be written using “-and”
- Response: The content indicated was corrected and indicated in the text with track changes.
- Conclusion could be elaborated as it is simply the replicate of abstract.
- Response: Conclusion was corrected.
Reviewer 2 Report
The manuscript requires major edition to be considered. Many manuscripts are in this field, then, the analysis for this topic should be deeper than in the current form.
The originality of the revised information should be highlighted to attract readers attention.
Minor mistakes are regarding grammar and typing.
Please check homogeneity of abbreviations (for example ROS for reactive oxigen species) and in the size of font.
Some chemical structures are required to enrich the understanding of putative actions of Ascorbate.
The content of subsections in pharmacology should be reoprganized. Some sentences seems to be adequate into other subsection of those included.
References should be analyzed, selected and changed. The most should be in the last 5 years as the topic has been explored, analyzed and revised many times in this period. Moreover, more than 200 articles are in pubmed with the terms 'ascorbic AND cancer' in each of the last 3 years. Many of them could enrich your observations from the preclinical approaches.
A carefully analysis of information from ClinicalTrial.gov must be done to enrich the content of your discussion and conclusions sections (at least 140 studies are available, some of them concluded; posology is particularly interesting when effectiveness is suggested or proved).
Author Response
Reviewer 2: The manuscript requires major edition to be considered. Many manuscripts are in this field, then, the analysis for this topic should be deeper than in the current form. The originality of the revised information should be highlighted to attract readers attention.
- Minor mistakes are regarding grammar and typing.
- Response: Grammar errors were corrected. In addition, the manuscript was sent to a professional English edition.
- Please check homogeneity of abbreviations (for example ROS for reactive oxigen species) and in the size of font.
- Response: The content indicated was corrected and indicated in the text with track changes.
- Some chemical structures are required to enrich the understanding of putative actions of Ascorbate.
- Response: The chemical structures of the different chemical forms of ascorbate were added within the equations for better understanding.
- The content of subsections in pharmacology should be reoprganized. Some sentences seems to be adequate into other subsection of those included.
- Response: A new structure was given to the subsections of pharmacology contents, including a better sequence order.
- References should be analyzed, selected and changed. The most should be in the last 5 years as the topic has been explored, analyzed and revised many times in this period. Moreover, more than 200 articles are in pubmed with the terms 'ascorbic AND cancer' in each of the last 3 years. Many of them could enrich your observations from the preclinical approaches.
- Response: The references were updated, and references up to 5 years old were added. In addition, information about the most recent studies on the use of ascorbate in cancer treatment, both in in vitro, in vivo and human models, was updated.
- A carefully analysis of information from ClinicalTrial.gov must be done to enrich the content of your discussion and conclusions sections (at least 140 studies are available, some of them concluded; posology is particularly interesting when effectiveness is suggested or proved).
- Response: A careful search was made about the latest ongoing studies on the use of ascorbate in cancer treatment, which was systematized in Table 2.
Reviewer 3 Report
In the paper “Ascorbate as a bioactive compound in cancer therapy: The old classic strikes back” González-Montero and co-workers reviewed the role of Ascorbic acid in cancer therapy.
In my opinion, this review is interesting, however there are diverse points of criticism that need attention, before being eligible for publication in Molecules.
1) The structure of the review is not well proportioned, because there is a large part that discusses Ascorbic acid in general (its historical origin, its use in medicine, its pharmacology and so on). However, since the review is focused on Ascorbate and cancer, I suggest shortening the general part and deepening the part inherent to cancer.
2) Introduction paragraph is devoid of references, please revise it.
3) In 2.8 section, before talking about “The ascorbate paradox: biphasic effect”, please add a comment on Hormesis.
4) Authors should revise Bibliography and references according to the Journal’s guidelines and by keeping the same style for both text and Bibliography section.
5) Please use “concentration” when talking about in vitro studies and “dose” for animal ones. In addition, both in vitro and in vivo expressions should be written in italics.
6) In section 3, Authors should resume each subsection with a table showing the reference, the experimental model employed and the pathways involved in order to help the reader easily seize what is described in the corresponding part.
7) There are also some grammatical, typing and punctuation errors, hence moderate English language and style editing is required.
Author Response
Reviewer 3:
In the paper “Ascorbate as a bioactive compound in cancer therapy: The old classic strikes back” González-Montero and co-workers reviewed the role of Ascorbic acid in cancer therapy. In my opinion, this review is interesting, however there are diverse points of criticism that need attention, before being eligible for publication in Molecules.
- The structure of the review is not well proportioned, because there is a large part that discusses Ascorbic acid in general (its historical origin, its use in medicine, its pharmacology and so on). However, since the review is focused on Ascorbate and cancer, I suggest shortening the general part and deepening the part inherent to cancer.
- Response: Critical information about recent in vitro, in vivo, and human studies of the effect of ascorbate in cancer treatment has been added and summarized in Tables 1 and 2.
- Introduction paragraph is devoid of references, please revise it.
- Response: We added references in the introduction paragraph, which is marked with track changes in the text.
- RR In 2.8 section, before talking about “The ascorbate paradox: biphasic effect”, please add a comment on Hormesis.
- Response: We added a full paragraph about Hormesis, for a better understanding of the biphasic effect of ascorbate.
- Authors should revise Bibliography and references according to the Journal’s guidelines and by keeping the same style for both text and Bibliography section.
- Response: References format was corrected according journal guidelines.
- Please use “concentration” when talking about in vitro studies and “dose” for animal ones. In addition, both in vitro and in vivo expressions should be written in italics.
- Response: The content indicated was corrected and indicated in the text with track changes.
- In section 3, Authors should resume each subsection with a table showing the reference, the experimental model employed and the pathways involved in order to help the reader easily seize what is described in the corresponding part.
- Response: We added two new tables, which summarize the effect of ascorbate in the treatment of cancer, both in in vitro models, in vivo and in human clinical studies (phase 1, phase 2 and phase 3).
- There are also some grammatical, typing and punctuation errors, hence moderate English language and style editing is required.
- Response: Grammar and syntax errors were corrected. In addition, as we are not native in English, the manuscript was subjected to a professional English edition. We are including the certificate of editing.
Round 2
Reviewer 2 Report
The comments and suggestion were considered. The manuscript is improved in this form.
Reviewer 3 Report
The Authors have responded adequately to all points.
However, both in the revised manuscript and in the supplementary file all tables are lacking. Therefore, it is suggested to revise all figures and tables before recommending this manuscript for publication in Molecules after minor revision.